# Sensory Appeal and Routines Beat Health Messages and Visibility Enhancements: Mixed-Methods Analysis of a Choice-Architecture Intervention in a Workplace Cafeteria

**DOI:** 10.3390/nu14183731

**Published:** 2022-09-10

**Authors:** Eeva Rantala, Elina Järvelä-Reijonen, Kati Pettersson, Janne Laine, Paula Vartiainen, Johanna Närväinen, Jussi Pihlajamäki, Kaisa Poutanen, Pilvikki Absetz, Leila Karhunen

**Affiliations:** 1VTT Technical Research Centre of Finland, 02044 Espoo, Finland; 2Institute of Public Health and Clinical Nutrition, University of Eastern Finland, 70211 Kuopio, Finland; 3Finnish Institute for Health and Welfare, 00271 Helsinki, Finland; 4Department of Medicine, Endocrinology and Clinical Nutrition, Kuopio University Hospital, 70029 Kuopio, Finland; 5Faculty of Social Sciences, Tampere University, 33520 Tampere, Finland

**Keywords:** choice architecture, workplace cafeteria, food choice, nutrition, health promotion, eye tracking, mixed methods

## Abstract

Easier recognition and enhanced visibility of healthy options supposedly increase healthy choices, but real-world evidence remains scarce. Addressing this knowledge gap, we promoted nutritionally favourable foods in a workplace cafeteria with three choice-architectural strategies—priming posters, point-of-choice nutrition labels, and improved product placement—and assessed their effects on visual attention, food choices, and food consumption. Additionally, we developed a method for analysing real-world eye-tracking data. The study followed a pretest–posttest design whereby control and intervention condition lasted five days each. We monitored visual attention (i.e., total number and duration of fixations) and food choices with eye tracking, interviewed customers about perceived influences on food choices, and measured cafeteria-level food consumption (g). Individual-level data represents 22 control and 19 intervention participants recruited at the cafeteria entrance. Cafeteria-level data represents food consumption during the trial (556/589 meals sold). Results indicated that the posters and labels captured participants’ visual attention (~13% of fixations on defined areas of interest before food choices), but the intervention had insignificant effects on visual attention to foods, on food choices, and on food consumption. Interviews revealed 17 perceived influences on food choices, the most common being sensory appeal, healthiness, and familiarity. To conclude, the intervention appeared capable of attracting visual attention, yet ineffective in increasing healthier eating. The developed method enabled a rigorous analysis of visual attention and food choices in a natural choice setting. We discuss ways to boost the impact of the intervention on behaviour, considering target groups’ motives. The work contributes with a unique, mixed-methods approach and a real-world setting that enabled a multi-dimensional effects evaluation with high external validity.

## 1. Introduction

Redesigning choice architectures—the way available options are presented in choice environments [1]—is a gentle, non-intrusive approach to promote healthy eating. The approach acknowledges people’s limited ability to regulate behaviour deliberately according to their self-declared interests [2], and seeks to facilitate healthy behaviours, inter alia, by making healthier options more effortless and visible [3]. For the working population, workplace cafeterias are regular food-choice environments that contribute to a substantial proportion of overall dietary intake [4,5]. Interventions conducted at workplace cafeterias have the potential to reach large audiences and improve workers’ nutrition and health [6], an outcome that benefits employers and the society as well [7,8].

Interventions that facilitate the recognition and enhance the visibility of health-promoting foods have proved capable of encouraging healthier eating [9]. Examples of such interventions include visual cues that prime for healthy choices [10], nutrition labels that communicate the nutritional properties of foods [11,12], and changes to product placement [13,14]. Primes can involve words or images that activate motivations for healthy eating, consequently enhancing people’s ability to recognize and choose healthy foods [15,16]. Nutrition labels prompt people to reassess their food choices at the point of choice and assist in identifying healthier options [17]. Improvements in product placement increase both visibility and convenience, for example, by placing healthy options at the eye level [18], first in line [19], or physically closer to the chooser [20]. Evidence suggests that the closer foods are the greater their consumption, and vice versa [14]. While priming and placement interventions influence behaviour more directly, often without noticing, nutrition labels require somewhat greater cognitive involvement [9,10]. Learnings from behavioural sciences stress the importance of conveying nutrition information in a way that considers actual human behaviour [16]. In food purchasing contexts, such behaviour typically follows decisions that build on simple-to-interpret cues rather than in-depth processing of detailed information [21,22]. According to the dual process theories of cognition, this translates to decision-making processes that employ automatic rather than reflective cognitive processes [23,24].

Despite a wealth of literature about priming, nutrition labels, and placement strategies, real-world evidence from workplace cafeterias remains limited and inconsistent [18,20,25,26,27,28,29]. Conflicting findings could be explained, for example, by the varying capabilities of interventions to capture participants’ visual attention, a prerequisite for strategies that influence via eyesight [9,30]. Alternatively, target populations’ diverse preferences could explain varying responses to interventions [31,32].

Attention is a limited resource that allows us to notice particular objects and decide whether to act upon them [33]. Attention can be captured by external stimuli that stand out in the visual field (i.e., bottom-up processing), or it can be driven by internal influences, such as prior experiences or current goals (i.e., top-down processing) [34,35,36]. Studies on visual attention have mainly relied on self-reports, such as interviews or questionnaires, although these methods may yield biased results, typically overestimating true attention [21,22,30,37].

Eye tracking enables the measurement of eye movements and provides an objective method to study visual attention and behaviour [38]. Fixations are eye movements that reflect exposure to visual stimuli [39]. During fixations, eyes hold gazed objects steady on the foveal region of the retina—the central 1–2 degrees of visual angle—enabling a detailed perception of fixated objects [40,41]. Since attention typically determines where the eyes go, fixations serve as proxies for the location of visual attention [38,39]. Visual attention, in turn, often projects the focus of current active processing [33]. In the food-choice context, greater visual attention may reflect the attraction of targeted foods [42] and predict subsequent food choices [15,36,43,44,45].

Eye tracking is commonly used in psychological, marketing, and consumer research, but it is less familiar to the fields of nutrition and health promotion. Moreover, the method has been applied nearly exclusively in hypothetical or simulated choice contexts [42]. While recent food-related eye-tracking experiments have moved from laboratories to more naturalistic environments [22,36,44,46,47], intervention studies are lacking that employ eye tracking to explore the food choice process in fully unconstrained real-world settings.

The aim of this study was to assess the effects of an intervention that promoted food choices of high nutritional quality in a workplace cafeteria with three choice-architectural strategies for easier recognition and enhanced visibility of targeted foods: (1) priming posters, (2) prominent point-of-choice nutrition labels, and (3) improved product placement. In addition, the study developed a method for analysing eye-tracking data collected in a natural choice setting. Quantitative effects evaluation considered individual-level visual attention (i.e., total number and duration of fixations) and food choices, as well as cafeteria-level food consumption. The hypotheses were that prominently displayed posters and labels would catch the eye, and that the intervention would increase visual attention to promoted foods as well as the choices and consumption of these foods. Qualitative analysis considered perceived influences on food choices, observations of the intervention, and understanding of the used nutrition label. The mixed-methods approach that employed objective and subjective data and integrated quantitative and qualitative elements [48] enabled a multi-dimensional examination of the cascade of intervention effects from perception to action.

## 2. Materials and Methods

### 2.1. Study Design

The study followed a quasi-experimental pretest–posttest design and took place in a workplace cafeteria in the Northern Savo region of Finland between January and February 2018. Pre-intervention measurements took place at baseline and served as control for post-intervention measurements that took place five weeks later, immediately after the launch of the intervention. Both measurement periods lasted five days (Monday through Friday), during which the cafeteria served identical menus. The study is an independent part of a larger type 2 diabetes prevention study, Stop Diabetes (StopDia) [49] reviewed by the research ethics committee of the hospital district of Northern Savo, Finland (statement number: 467/2016, date of approval: 3 January 2017).

### 2.2. Setting

The study cafeteria was located in a municipal office building in an urban area and served approximately 150 customers per day. The clientele consisted predominantly of employees of the office and nearby workplaces, yet also included some individuals outside the working life, such as senior citizens and students. Lunch hours were daily from 10.30 a.m. to 1 p.m.

The cafeteria operated in a self-service model in which customers choose and compose their meals from a serving line (Figure 1). The cafeteria provided daily four warm main course options: two fish/meat courses, one vegetarian course, and one soup, together with relevant carbohydrate accompaniments (rice and/or potatoes) and steamed vegetables (see Appendix A for the entire menu). The warm meals also included bread, beverages, and side salad. In addition, the cafeteria provided a salad bar as a cold main course option. The salad bar consisted of 18−19 salad components per day, including vegetables and fruits, mixed salads, protein sources (meat, egg, cheese, pulses, fish, and tofu varieties), and condiments (seeds, nuts, tortilla chips, and roasted onion crumbs), as well as a variety of dressings. Furthermore, the cafeteria sold some snacks and desserts. The cafeteria and its food offering represented a typical workplace cafeteria in Finland.

The cafeteria belonged to the Heart-symbol system of the Finnish Heart Association and the Finnish Diabetes Association (www.sydanmerkki.fi/en, accessed on 9 September 2022). The Heart symbol (Figure 2) is a voluntary, positive nutrition label that EU-Regulation (EC No. 1924/2006) acknowledges as a nutritional claim, and that food manufacturers and caterers can apply for their products. The symbol indicates nutritionally better choices that meet product category-specific nutrition criteria for the quantity and quality of fat, and the quantity of salt, sugar, and fibre. The criteria build on the Finnish nutrition recommendations [50], acknowledge major public health nutrition challenges prevalent in Finland [4], and are updated regularly by an independent expert group that consist of professionals in nutrition and medicine [51]. Since its launch in 2000, the symbol has become familiar to the majority of Finnish adults [51]. As a member of the Heart-symbol system, the study cafeteria was committed to provide daily at least one option in defined product categories (main course, side dish, bread, fat spread, milk/sour milk, salad, and salad dressing) that fulfilled the Heart-symbol criteria. Yet, the cafeteria was allowed to also provide options that did not meet the criteria. The cafeteria had standardised recipes with calculations of the nutritional content of all foods prepared in the kitchen, and knowledge on the food items in their offering that met and did not meet relevant Heart-symbol criteria. Additionally, the Finnish Heart Association had granted the cafeteria the right to label their criteria-fulfilling foods with the Heart symbol. At baseline, however, the cafeteria did not inform their customers about the Heart symbol nor indicate corresponding options on the serving line (Section 2.4). Hence, while the cafeteria was a member of the Heart-symbol system and offered corresponding options, the customers had no way of knowing this merely by observing the cafeteria environment. The study cafeteria provided thus an appropriate setting to study the effects of choice-architectural strategies that facilitate the recognition and enhance the visibility of nutritionally beneficial options while the food offering remained unchanged. Hereafter, we refer to options that meet the nutrition criteria of the Heart symbol as “heart-foods” and options that do not meet the criteria as “non-heart-foods”.

### 2.3. Participants and Their Recruitment

A week before each study period, we displayed notices in the cafeteria that informed of a forthcoming consumer study and the opportunity for customers to participate. The personnel of the office building that housed the cafeteria received this information also via email from their human resources manager. During the study periods, we recruited participants at the entrance of the cafeteria. Participants were informed about study aims (“customer perceptions in the cafeteria”) and data collection methodology. Participation in eye tracking required the ability to navigate through the cafeteria and to compose and pay for the lunch without eyeglasses. Hence, customers could participate either in eye tracking and interview or in interview only. During the intervention, to maximise data collected, we allowed participation both for customers that had participated during the control condition and for customers that had not. As a result, the samples recruited during control and during intervention included partly same and partly different individuals (Table 1). We thus apply statistical methods developed for comparing two partially overlapping samples that include both paired and independent observations [52,53,54,55]. Participants received fruit as a compensation for their contribution.

Altogether, 41 customers (control: *n* = 22, intervention: *n* = 19) participated in eye tracking and interviews, and an additional 51 customers (control: *n* = 30, intervention: *n* = 21) only in interviews. In this paper, we report the results of the sample that participated in both eye tracking and interviews because the data collected from the interview-only sample did not yield significant additional information relevant to the research questions of the current study (data not shown). The included sample of the control condition comprised 14 (64%) men and 8 (36%) women with a mean age of 43 years (SD 12, range 19–63). The included sample of the intervention condition comprised 10 (53%) men and 9 (47%) women with a mean age of 46 years (SD 10, range 31–63). No significant between-condition differences were found in the distributions of gender (partially overlapping samples “*z*_8_”-test for comparing proportions [52]: statistic = −0.879, *p* = 0.379) or age (partially overlapping samples *t*-test “*T*_new1_” with equal variances assumed [53,55]: statistic = −0.986, *p* = 0.332). The intervention sample included eight individuals (six men and two women with a mean age of 44 years [SD 9, range 31–58]) who had participated also during the control condition in eye tracking and interviews (*n* = 7) or only in interviews (*n* = 1). The gender and age distribution of these participants did not differ significantly from other participants of the intervention condition (Fisher’s exact test *p* = 0.170; *t* [17] = −0.770, *p* = 0.452, respectively).

The majority of participants appeared to be familiar with the cafeteria. Although we did not enquire about this specifically, 11 (50%) participants of the control condition clearly implied in their reports that they had been in the cafeteria before. On the other hand, only one participant of the control condition mentioned not having been to the cafeteria before. In the intervention condition, all participants were assumed to be familiar with the cafeteria because none declared themselves first-timers when we asked if they noticed any changes in the cafeteria compared to earlier.

### 2.4. Control and Intervention Condition

During the control condition, we made no changes in the cafeteria. At this point, Heart-foods were not readily identifiable on the serving line. Their recognition required efforts to search for nutrition information typically provided in the small print of menus or product packages (Table 2). The arrangement of heart-food and non-heart-food options on the serving line was not systematic.

During the intervention, we promoted heart-foods with three choice-architectural strategies: priming, point-of-choice nutrition labels, and placement (Table 2). The first author (E.R.) was responsible for the implementation, made needed adjustments each day before the beginning of the lunch service, and monitored the quality of implementation throughout the intervention. The priming strategy displayed posters at the cafeteria entrance and on the serving line (Figure 3), and the point-of-choice labelling strategy indicated all available heart-foods with Heart symbols (Figure 4). The only exception was the salad bar whereby limited space impeded labelling individual salad components separately and unambiguously. The salad components were hence labelled as a whole, and a sign informed that the salad bar enables composing a meal that deserves the Heart symbol. Consequently, our data analyses categorised all salad components as heart-foods. Salad dressings, however, were labelled individually.

The placement strategy set heart-food items first in line and towards the front row and non-heart-food items last in line and towards the back row within product categories (Table 2). Heart-food snack items were additionally lifted on the top shelf of a display to enable eye-level view. On serving line stretch #2 (Figure 1), the placement manipulation excluded the soup whose position was fixed due to the layout of the serving line. In addition, according to the wishes of the cafeteria staff, the placement of a couple of other warm courses remained suboptimal on two days of the intervention week due to practicalities concerning cleanliness and food sufficiency. Otherwise, the implementation on stretches #1–2 followed plans. According to literature, expecting perfect or near-perfect implementation is unrealistic and unnecessary because few interventions have reached implementation levels closer than 80% of optimal and because studies have yielded positive results with levels around 60% [56]. We hence considered the implementation on stretches #1–2 overall satisfactory. On serving line stretch #3, however, the implementation faced major challenges throughout the intervention week because most food items kept travelling away from their assigned places and corresponding Heart labels as customers handled them. Such implementation quality was unacceptable, as the findings would not have reflected the intended intervention. Hence, our data analyses considered only data collected at stretches #1–2.

### 2.5. Data Collection

Our data collection methods comprised eye tracking, recording cafeteria-level food consumption, and interviews. Collected data involved no identifiable information on study participants.

#### 2.5.1. Eye Tracking

We collected eye-tracking data to study the effect of the intervention on visual attention to Heart-symbol materials, heart-foods, and non-heart-foods, as well as on food choices. The recording started before participants reached the beginning of the serving line and ended after they left the serving line (Figure 1). The data were collected with video-based mobile eye-tracking glasses (iViewETG 2.7, SensoMotoric Instruments GmbH, Teltow, Germany) that take 30 frames per second (i.e., 30 Hz binocular sampling rate) and feature a scene camera with a resolution of 1280 × 960 pixels (Figure 5). This device captures the wearer’s eye movements with two small cameras on the bottom rim of the glasses and maps the point of gaze into a scene video [57]. An experienced research technician was responsible for handling the eye-tracking device throughout the study.

We calibrated the eye-tracking glasses for each participant with a three-point calibration protocol [57] (Figure 5a). After the calibration, we added study identification codes on participants’ trays and instructed participants to proceed to the serving line, compose the lunch meal of their choice, and pay for the meal as they normally would (Figure 5b). After leaving the serving line, the research technician took the eye-tracking glasses and guided participants to the interview (Section 2.5.3).

#### 2.5.2. Cafeteria-Level Food Consumption

To compute cafeteria-level food consumption, we manually recorded the weights (g) of all food items available on the serving line during the lunch service, as well as corresponding leftovers at the end of the service. This procedure recurred every day during control and intervention. Before the beginning of the lunch service, we obtained recorded measures by weighing served food items with the cafeteria’s kitchen scale, by consulting waybills that reported the quantities of foods supplied from the caterer’s central kitchen, and/or by information that manufacturers provided on packaged food products. During the lunch service, the cafeteria staff reported the type and quantity of foods they added on the serving line. At the end of the service, we weighed all leftovers with the same kitchen scale as before the beginning of the service. The consumption data were recorded by the first and the second author (E.R. and E.J.-R.) together with three nutrition students.

#### 2.5.3. Interviews

After participants had composed their meals and before they started to eat, we photographed their trays and interviewed them. The interview aimed to capture factors participants perceived to influence their food choices, as well as participants’ observations of the intervention and understanding of the Heart symbol. During both control and intervention, we enquired perceived influences on food choices with three questions: one about factors participants paid attention to on the serving line while composing meals, one about factors that determined participants’ choices on the participation day, and one about factors participants usually held to be important when choosing foods. In addition, we asked whether participants’ choices on the participation day were typical of them. Recorded demographics comprised age and gender.

During the intervention, we additionally asked whether participants noticed any changes in the cafeteria. If they did, we asked them to elaborate the observed changes, their opinion on the changes, as well as perceived effects of the changes on their food choices. At the end of the interview, we showed participants the Heart symbol (Figure 2) and asked if they were familiar with it and how did they interpret it. Finally, participants reported whether they had participated in the study also during the control condition.

The interviews lasted up to five minutes per participant and were conducted by the second author—an authorised nutritionist (E.J.-R.)—together with two nutrition students. Longer interviews were not feasible, because participants had to be dismissed before their foods got cold. The interviewers took field notes of participants’ answers and typed the answers as soon as possible after the interviews.

### 2.6. Analyses

#### 2.6.1. Fixations on Heart-Symbol Materials and Foods

We analysed the collected eye-tracking data with SMI BeGaze^TM^ 3.4 behavioural and gaze analysis software build 52, 2014© [58]. This software detects fixations with a dispersion-based algorithm that identifies fixations as groups of consecutive data points within a particular dispersion [59]. The software uses a minimum fixation-duration threshold of 80 ms. We analysed the detected fixations with a scan path visualisation that indicates the point of each fixation with a colourful circle on the scene video that represents participants’ field of vision (Figure 4). In this visualisation, the software uses a maximum fixation-duration threshold of 500 ms. Following the software thresholds, we limited our analysis to 80–500 ms long fixations. The analysis covered the section between participants’ arrival to serving line stretch #1 and the moment when they had passed by the targets of the intervention at stretch #2 (Figure 1). This section had satisfactory implementation throughout the intervention. We exported full reports of each participant’s fixations from the eye-tracking data analysis software, and from these reports, extracted 80–500 ms long fixations within the target section.

We coded the extracted fixations manually based on visual inspection of freeze-frames from participants’ eye-tracking videos; a method used in coding eye-tracking data from shopping environments [22,44]. The coded data comprised in total 7261 fixations (control: *n* = 3581, intervention: *n* = 3680) from 37 participants with unbroken eye-tracking recordings (control: *n* = 19, intervention: *n* = 18). The recordings of two participants (control: *n* = 1, intervention: *n* = 1) appeared to be poorly calibrated, however, and were excluded from the analysis. After this exclusion, the data covered 6949 fixations (control: *n* = 3368, intervention: *n* = 3581) from 35 participants with successful eye-tracking recordings (control: *n* = 18, intervention: *n* = 17).

The first author (E.R.) was responsible for coding the fixations. She had been involved in designing and implementing the intervention, and she knew the locations of all objects of interest on the serving line as well as the categorisation of available foods into heart- and non-heart options. Such single-coder approach has been used in eye-tracking research with manually laborious analysis [46], and is a methodologically sound choice as long as it includes checks on validity and reliability [60]. We promoted validity and reliability through a peer-checking process typical of qualitative research [60,61]. The peer-checking meant that the first author iteratively reviewed samples of fixations and suggested coding with several other authors (E.J.-R., K.P. (Kati Pettersson), J.L., J.N., P.A., and L.K.), and the authors discussed, refined, and agreed on the coding.

For the coding, we listed all objects of interest on the serving line stretches #1–2 (Appendix A) and defined the area of interest (AoI) for each object (Appendix A). The AoIs covered the priming posters and point-of-choice Heart symbols added on the serving line during the intervention, as well as all heart-food and non-heart-food items available during control and intervention. We coded fixations whose point indicators touched any AoIs in the video frame (control: *n* = 1489 [44.2%], intervention: *n* = 1761 [49.2%]) according to three target groups: (1) Heart-symbol materials, (2) heart-foods, and (3) non-heart-foods. The coding was not mutually exclusive, because the fixation point indicator could touch several objects simultaneously. In such situations (control: *n* = 24 [0.7%], intervention: *n* = 206 [5.8%]), fixations received codes according to all touched objects. If fixation targets were unidentifiable due to long distance and/or blurry video, fixations were excluded and coded “unclear*”* (control: *n* = 1 [0.03%], intervention: *n* = 3 [0.08%]). Fixations that touched foods on participants’ own or other customers’ plates were coded according to their targets only when the plates and hence the fixated foods were lifted over corresponding AoIs on the serving line during portioning, and before the point of choice of the foods were passed (control: *n* = 116 [3.4%], intervention: *n* = 130 [3.6%]).

Besides coding the fixations that touched AoIs according to their targets, we also coded these fixations depending on their timing relative to food choices. A food choice referred to the first time when participants started to portion a given food. Moments of choice were determined case-by-case and involved, for example, reaches for food items or their serving utensils, reaches for salad bowls reserved for customers that chose the salad bar, or moments in which participants began to remove the caps of salad dressing bottles to enable pouring. Since the study aims to capture the potential effect of the intervention on food choices, we were particularly interested in fixations that preceded food choices. We gave fixations a code “pre” when they touched foods or related Heart-symbol materials before choices were made concerning the targeted foods (control: *n* = 674 [20.0%], intervention: *n* = 991 [27.7%]), and focus further analyses on these fixations. This coding was conducted at the level of food item, except for salad components in the salad bar that were considered as a whole (Appendix A).

At serving line stretch #1 (Figure 1), the intervention comprised one “Follow the heart”-poster (Figure 3) and one Heart symbol (Figure 2) attached to a salad bar notice. Only two participants of the intervention condition had fixations that swept the AoIs of these objects (2–3 fixations per participant). Hence, we chose to limit further analyses to fixations at serving line stretch #2, for which most of the AoIs were drawn and whereby participants made their actual food choices. The final data set comprises 1660 fixations (control: *n* = 674, intervention: *n* = 986) on AoIs before food choices were made. Within this sample, the proportion of fixations with overlapping targets (Heart-symbol materials, heart-foods, and/or non-heart-foods) is 12.0% (control 3.4%, intervention 17.8%).

Our main outcome measures are the total number and total duration of fixations participants had on Heart-symbol materials, heart-foods, or non-heart-foods before food choices at serving line stretch #2. Due to between-participant differences in the time spent at the serving line and in the number of fixations accumulated during this time, we follow a procedure used before [15,46] and report the outcomes as the percentages of participants’ total fixations on AoIs before food choices at stretch #2. The final study sample consists of 17 participants of the control condition and 17 participants of the intervention condition, excluding one participant of the control condition who had zero fixations on AoIs, and to whom we were thus unable to compute percentages. In addition to the main outcomes, to illustrate the share of overall visual attention that fixations on AoIs covered, we report the percentages of these fixations within participants’ total fixations by the analysed section of serving line stretch #2. We examined differences between the control and the intervention condition using the partially overlapping samples *t*-test “*T*_new1_” for comparing the means of normally distributed variables with equal variances [53,55], and the non-parametric counterpart “*T*_RNK1_” test for assessing the location shift of non-normally distributed variables with equal variances [54]. We checked the normality assumption with the Shapiro-Wilk test and the visual inspection of distribution curves, and the equality of variances assumption with the partially overlapping samples variances test “*T*_var1_” [62]. We report all *p*-values two-tailed, using *p*-value 0.05 as the level of statistical significance. For data management and analysis, we employed Microsoft Excel^®^ 2016 (Redmond, WA, USA), IBM SPSS^®^ Statistics 28.0 (Armonk, NY, USA), and R version 4.2.1 [63] with the “Partiallyoverlapping” R-package version 2.0 [64].

#### 2.6.2. Food Choices

We tracked participants’ food choices from their eye-tracking videos and recorded each food item participants added on their trays. With four participants whose eye tracking failed entirely so that the recordings could not be played (control: *n* = 3, intervention: *n* = 1), we relied on their interview answers and photos taken of their meals (Section 2.5.3). We examined food choices at the level of food item, considering individual snacks, salad components, salad dressings, warm courses, condiments, and desserts chosen from serving line stretch #2 (Appendix A).

Our main outcome measures are the number of food items chosen per participant during control and intervention, and the percentages of these items that were heart- and non-heart-options. As the outcome variables did not follow a normal distribution across the conditions (Shapiro-Wilk test *p* < 0.05), we examined differences between the control and the intervention condition using the non-parametric partially overlapping samples “*T*_RNK1_” test with equal variances assumed [54]. We checked the equality of variances assumption with the partially overlapping samples variances test “*T*_var1_” [62]. We report all *p*-values two-tailed, using *p*-value 0.05 as the level of statistical significance. We ran the analyses with and without participants who chose the salad bar as they had a greater number of items to choose from compared to warm-course choosers, and because all salad components were categorised as heart-food items (Section 2.4). For data management and analysis, we employed Microsoft Excel^®^ 2016 (Redmond, WA, USA), IBM SPSS^®^ Statistics 28.0 (Armonk, NY, USA), and R version 4.2.1 [63] with the “Partiallyoverlapping” R-package version 2.0 [64].

#### 2.6.3. Cafeteria-Level Food Consumption

To obtain cafeteria-level estimates of food consumption, we subtracted the weight (g) of leftovers from the weight of foods served over the lunch service. The analysis covered food items available on serving line stretch #2 (Figure 1, Appendix A), excluding snacks and desserts due to incomplete data collection. Among the foods included in the analysis, missing data concerned 0.56% of total measurements. With food items that were available daily, missing measurements were replaced with the mean consumption of the given food item during the rest of the given study condition. With food items that were not available every day, missing data led to the removal of the items from the control and the intervention data of the given weekday. Our main outcome measures are the total volume of foods consumed (g) during control and intervention divided by the number of meals sold over each period, and the percentages of these consumption volumes that heart-foods and non-heart-foods covered. Similar to the food-choice analysis, we report the consumption results with and without meals composed from the salad bar. For data management and analysis, we employed Microsoft Excel^®^ 2016 (Redmond, WA, USA).

#### 2.6.4. Perceived Influences on Food Choices

We employed descriptive qualitative content analysis [65] to identify and code factors participants perceived to influence their food choices. We employed a coding matrix that built on the nine dimensions of the Food Choice Questionnaire (FCQ) that assesses perceived influences on food selection at the individual level [66]. The tool has proved applicable across cultures and populations [67]. The nine dimensions of the FCQ are: health, mood, convenience, sensory appeal, natural content, price, weight control, familiarity, and ethical concern. In addition, we included in our coding matrix the dimension “openness to experience” from the NEO Personality Inventory (NEO-PI) [68]. This personality trait predicts willingness to try new foods [69], and has proved to correlate negatively with the FCQ factor “familiarity” [66]. For data management and analysis, we employed NVivo R1.6 (QRS International) and Microsoft Excel^®^ 2016 (Redmond, WA, USA).

The first author (E.R.) systematically coded the data according to the coding matrix, maintaining the freedom to modify category headings to reflect the content of the interview data better. For example, the NEO-PI dimension “openness to experience” evolved into “variation”. When relevant, new categories were generated following the principles of inductive qualitative content analysis [65]. The coding was not mutually exclusive, meaning that individual interview answers could receive several codes. The validity and reliability of the coding was ensured with a peer-checking method common in qualitative research [60,61]. The first author reviewed example quotes from the interviews against suggested coding with the second and the last author (E.J.-R. and L.K.), and the three authors discussed, refined, and agreed on the coding. We portray identified influences narratively and report the number of individuals that mentioned each influence during control, during intervention, and altogether.

#### 2.6.5. Self-Reported Observations and Understanding of the Heart Symbol

Regarding observations of the intervention and understanding of the Heart symbol, we report the number of participants in the intervention group that identified the intervention, and the number of participants that were familiar with and correctly understood the Heart symbol.

## 3. Results

### 3.1. Fixations on Heart-Symbol Materials and Foods

The median time that participants spent at the analysed section of serving line stretch #2 was 40 s (interquartile range [IQR] 37 s, range 17−126 s) in the control condition (*n* = 17) and 55 s (IQR 40 s, range 9–220 s) in the intervention condition (*n* = 17). The difference between the conditions was not statistically significant (*T*_RNK1_ = −0.499, *p =* 0.622). Within this time, participants accumulated a median of 103 (IQR 61, range 49–353) fixations during control and 141 (IQR 81, range 11–517) fixations during intervention (*T*_RNK1_ = −0.667, *p =* 0.511). Of these fixations, the median proportion that fell on the defined areas of interest (AoI, Appendix A) before food choices was 34.0% (IQR 25.8%, range 5.7–68.5%) during control and 37.5% (IQR 23.2%, range 5.7–68.1%) during intervention (*T*_RNK1_ = −0.995, *p =* 0.329) (Figure 6a). These proportions, respectively, covered a median of 27.9% (IQR 28.2%, range 5.2–69.6%) and 37.8% (IQR 22.4%, range 4.8–69.2%) of the total duration of analysed fixations (*T*_RNK1_ = −1.071, *p* = 0.294) (Figure 6b). In absolute terms, before making their food choices, participants gazed at the AoIs for a median of 30 (IQR 36, range 17–89) fixations during control and for 52 (IQR 52, range 6–205) fixations during intervention (*T*_RNK1_ = −1.172, *p* = 0.252). The median total duration of these fixations was 4.7 s (IQR 6.7 s, range 2.8–17.4 s) during control and 10.0 s (IQR 10.2 s, range 0.8–41.3 s) during intervention (*T*_RNK1_ = −1.294, *p* = 0.207).

During the intervention, fixations on Heart-symbol materials covered on average 12.9% (SD 7.5%, range 3.8–27.3%) of the total number and 13.5% (SD 7.4%, range 4.2–27.9%) of the total duration of fixations on AoIs before food choices at serving line stretch #2 (Table 3). Regarding the percentage of fixations on heart-foods, the mean differences between intervention and control were not statistically significant for fixation number (*T*_new1_ = 0.387, *p* = 0.702) or duration (*T*_new1_ = 0.406, *p* = 0.688). The same applied to the number (*T*_new1_ = −0.706, *p* = 0.486) and duration (*T*_new1_ = −0.726, *p* = 0.474) of fixations on non-heart-foods.

### 3.2. Food Choices

The food-choice analysis considered all food items chosen from serving line stretch #2. However, the results reflect nearly exclusively participants’ main course and condiment choices because no participant purchased a dessert and only one participant purchased a snack to accompany their lunch. Participants chose a median of three (range 1–10) food items during control (*n* = 22) and three (range 1–13) items during intervention (*n* = 19) with no statistically significant difference between the conditions (*T*_RNK1_ = 0.075, *p* = 0.941) (Table 4). Of these choices, the median percentage of heart-food items was 33% (range 0–100%) during control and 67% (range 0–100%) during intervention. The change from control to intervention was not statistically significant (*T*_RNK1_ = −1.149, *p* = 0.261). Vice versa, the median percentage of non-heart-food items chosen was 67% (range 0–100%) during control and 33% (range 0–100%) during intervention (*T*_RNK1_ = 1.149, *p* = 0.261). The results did not change significantly after the exclusion of participants who chose the salad bar (control: *n* = 2, intervention: *n* = 3) (Table 4).

### 3.3. Cafeteria-Level Food Consumption

The cafeteria-level consumption analysis covered food items consumed from serving line stretch #2, except for snacks and desserts. Hence, similar to the food-choice results, the consumption results reflect main course and condiment consumption, corresponding to 556 meals sold during control and 589 meals sold during intervention. The overall amount of food consumed per sold meal was 15 g smaller during intervention compared to control (Figure 7a). Yet, between-condition differences in the percentages of heart-foods and non-heart-foods consumed were negligible. The percentage of heart-foods consumed was approximately 45% and the percentage of non-heart-foods approximately 55% during both study conditions (Figure 7b). Excluding the consumption of salad bar items, which corresponds to 68 (12.2%) meals sold during control and 76 (12.9%) during intervention, the overall amount of food consumed per sold meal was 24 g smaller during intervention compared to control (Figure 7c). The proportion of heart-foods consumed reduced from 40% during control to 38% during intervention, and the share of non-heart-foods consumed increased from 60% to 62% (Figure 7d).

### 3.4. Perceived Influences on Food Choices

We identified 17 factors participants perceived to influence their food choices (Table 5). The most frequently mentioned influence was sensory appeal, followed by healthiness, familiarity, and particular foods. Participants often reported multiple influences, and the decisive influence could depend on the choice task. For example, sensory appeal could determine individual food items chosen, while health considerations guided meal composition and portion size. We portray the identified influences briefly in a descending order according to the total number of individuals that mentioned each influence. Appendix A provides example quotes that reflect each influence.

#### 3.4.1. Sensory Appeal, Healthiness, Familiarity, and Particular Foods

Influences related to sensory appeal encompassed the look, taste, and texture of food. In addition, sensory appeal covered less-specified preferences that appeared in liking or disliking, wanting or not wanting, or finding foods tempting or not tempting. For several participants, sensory appeal was a priority that could outweigh competing influences such as healthiness. Highlighting the importance of taste, one participant said that if available foods were not appealing, they would go and eat elsewhere, even if it was more time-consuming and expensive.

Influences related to healthiness covered general, less-specified preferences for healthy choices, as well as considerations of meal composition, nutritional content, specific dietary guidelines, and the Heart symbol. Regarding meal composition, many participants focused on the proportion of vegetables and/or protein sources on the plate, and mentioned following the so-called plate model. In this model, vegetables fill half of the plate, protein-rich foods a quarter, and carbohydrate-rich foods another quarter. Considerations of nutritional content focused on protein, micronutrients, or the quality of fat. A few participants were motivated by national food-based dietary guidelines that recommend eating fish 2–3 times per week and a handful (i.e., 30 g) of nuts and seeds daily [50]. One participant was accustomed to use the Heart symbol to support their food choices.

Familiarity appeared in habitualness, in familiar choices that built on earlier experiences, and in preferences for traditional foods. Expressions that reflected habitualness included “always”, “daily”, “often”, and “usually”. Participants could, for example, “have a warm meal daily”, “often choose the salad bar”, or favour “foods they usually eat”. Habitualness manifested itself also in principles that guided participants’ choices such as an aim to include salad in the meal or a routine to choose specific courses whenever they are on the menu. Habits appeared to influence the choices of the majority of participants, as 95% of participants in the control condition and 84% in the intervention condition considered their choices on the participation day typical or somewhat typical of them.

Particular foods or food groups that drove participants’ choices included vegetables, fish, meat, bread, and soup. A number of participants considered important to include vegetables in the meal, and some favoured fish courses when they were available. These preferences relate to the healthiness- and familiarity-related influences concerning meal composition, dietary guidelines, and habitualness.

#### 3.4.2. Variation, Weight Control, and Menu

The importance of variation related to a desire to have a wide variety of options to choose from and a desire to choose diverse foods. A wide variety of salad components in the salad bar, for example, could prompt a decision to have salad. Seeking variety appeared also in a desire to choose foods different to those eaten elsewhere, in a curiosity to try new foods, and in a motivation towards specialties rarely served.

Factors related to weight control differed from factors related to healthiness in a more pronounced focus on weight management, on lightness, and/or on the conscious regulation of portion sizes. Participants mentioned balancing their eating with energy consumption and expressed preferences for options with low energy and/or fat content. The conscious reflection of portion sizes supported attempts to downsize portions and served as a means to compensate food choices considered less favourable. For example, participants could choose hamburgers yet omit potato wedges to keep the meal light and portion sizes reasonable.

The menu, which was available online, at the cafeteria entrance, and on the serving line, could determine both the restaurant in which participants chose to have lunch and the main courses they chose to eat. Participants could make their main course choices based on the menu without looking at the foods on the serving line. The menu relates to sensory appeal and familiarity, as participants could consider menu items by imagining their sensory properties and by recalling earlier experiences on similar foods.

#### 3.4.3. Further Factors

Further factors included considerations of satiety and mood. Prioritising satiety meant choosing foods that fill the stomach and take away the hunger. Participants could favour heartier foods, such as sausages or steak if they were very hungry or had a long day ahead. Relatedly, considerations of mood appeared in a preference for foods that help stay awake and cope with work commitments and leisure activities. Alternatively, mood could mean choosing foods based on current “vibes”.

Special dietary requirements, such as gluten free or vegetarian diet guided the choices of some participants, and a few participants paid attention to food quality such as freshness. Quality appraisals relate to sensory appeal because judgements of quality often built on sensory evaluation. Convenience was important for a few participants who preferred foods that are quick to acquire or eat. A few participants valued affordable prices and prices to quality ratio. For some, the season influenced food choices, as New Year’s resolutions motivated increased vegetable consumption, and cold weather prompted to choose warm foods. Social influences worked through the recommendations of the cafeteria staff or the experiences of other customers. Natural content reflected preferences for less processed foods, and ethical concerns focused on food origin.

### 3.5. Self-Reported Observations and Understanding of the Heart Symbol

During the intervention, two participants (11%) reported that they noticed changes in the cafeteria and correctly specified the changes as the point-of-choice Heart symbols. Both participants considered the symbols a positive add, and one of them said that the symbols influenced their choices. This person was used to paying attention to and consulting Heart symbols when choosing foods. Three additional participants (16%) remembered that they noticed the symbols after the interviewer showed them the symbol and asked whether it was familiar. No participant mentioned having noticed the priming posters or changes in the placement of foods. Nearly all participants (*n* = 17 [89%]) were familiar with the Heart symbol, and all participants understood the label to indicate healthier or nutritionally beneficial foods. Participants associated the label with healthy, heart-friendly, lighter, and/or nutritionally wiser foods with better salt and/or fat profile.

## 4. Discussion

This study used a unique mixed-methods approach to examine the effects of a multi-strategy choice-architectural intervention in a workplace cafeteria, and developed a method to analyse eye-tracking data collected in a natural choice setting. Three intervention strategies—priming posters, point-of-choice nutrition labels, and enhanced product placement—proved capable of capturing customers’ visual attention to the posters and labels but had no significant effects on visual attention to foods, on food choices, or on food consumption. Although health considerations influenced the food choices of a substantial proportion of participants, health-related motives were challenged by numerous competing priorities—particularly sensory appeal and familiarity.

### 4.1. Fixations on Heart-Symbol Materials and Foods

While few participants recalled having noticed the Heart-symbol materials, our eye-tracking data indicated that before making their food choices all participants gazed the materials at least once. Fixations that swept Heart-symbol materials covered on average 13% of fixations that targeted defined areas of interest before food choices. These findings suggest that the materials were sufficiently prominent to catch the eye and support evidence according to which prominent display, larger size, and distinctive colours enhance the noticing of nutrition labels [42,45,70]. In addition, our findings align with the conception that self-reports may yield inaccurate estimates of visual attention [21,30,37], and that eye tracking yields more accurate estimations of visual experience [22].

Gazing the Heart-symbol materials, however, does not mean that participants consciously paid attention to the materials or internalised their message [21,33], which assumingly leads to stronger effects on behaviour [30]. While fixations give a good estimation of visual attention and cognitive processing in some situations, we acknowledge that this may not always be the case because the direction of gaze may dissociate from the focus of attention [38], and because exposures to visual cues may occur by sheer accident [21,30]. This means that participants may have looked at the Heart-symbol materials while thinking of something else. To influence behaviour, exposure to nutrition labels must be accompanied with the perception and understanding of the label information [30].

In the case of the Heart symbol, however, the message is very simple, and according to our interviews, all participants understood the symbol more or less correctly. The symbol’s simple graphical layout also enables grasping the message quickly at a glance, particularly if the symbol is familiar, which was the case for nearly all participants. According to research on nutrition labels, simple graphic presentations and summary indicators are cognitively quicker and easier to process compared to numerical information and multidimensional label formats that consist of more than one piece of nutritional information [37,43,71,72,73]. Since familiarisation tends to reduce visual attention and response time to visual cues [35,36,74], and since visual cues can be perceived with the peripheral vision as well [38,47], our participants were likely able to perceive the Heart symbols quickly, even without looking at them directly. Hence, lack of understanding of the symbol is an unlikely explanation to the ineffectiveness of the intervention. A more presumable reason is that the message the symbol conveys was personally not relevant enough for the participants to overrule other simultaneous drivers of food choice [75], and that they chose to ignore the symbols [21]. This interpretation aligns with the finding that while many understand nutrition information, fewer actually use it, likely due to lack of motivation [76]. Consumers may consider, for example, that the Heart symbol is more relevant for individuals with or at risk of cardiovascular diseases, or that heart-foods are less tasty compared to non-heart-foods—a conception that may coexist with health consciousness [77].

### 4.2. Food Choices and Consumption

Our results mirror the findings of some point-of-choice labelling interventions that encouraged healthier food choices in workplace and university cafeterias with symbol-type nutrition labels and supporting communications material [26,27,29]. The Choices nutrition logo had negligible effects on the sales of fruit, healthier sandwiches, and soups in two workplace cafeterias [29], and a star-rating intervention proved ineffective in improving meal choices and nutrient intake in a university cafeteria [26]. Similarly, a lightning bolt symbol accompanied with calorie, fat, and cholesterol information had no effect on the sales of low-fat main courses in a military dining facility [27].

On the other hand, a communications campaign that was tailored to customers’ motivations was able to increase moderately healthy food choices and the total number of meal components chosen per participant in a military dining facility [25]. This intervention employed point-of-choice labels, posters, and floor stickers with slogans, such as “GO LEAN” and “GO FRESH” that reflected the military personnel’s desire to eat well to support performance. Complementing the intervention with a placement strategy that moved healthy options to more prominent and convenient places further improved the results [25]. Similarly, another intervention with point-of-choice nutrition labels, related communications material, and enhanced placement succeeded in increasing the sales of healthier items and in decreasing the sales of less healthy items in a hospital cafeteria [18]. In this study, the health-focused context may have supported intervention effectiveness, since hospital staff and patients might be particularly responsive to messages that encourage healthy eating. In summary, these findings suggest that a tailored approach is advisable in choice-architectural interventions. The conclusion receives support also from other recent studies [78,79].

The ineffectiveness of our placement intervention may be due to relatively minor changes in the order and physical distance of healthier options. Despite the rearrangement, all food items that our analyses covered remained fairly effortless to access and stayed on participant’s route to the cash desk. Although a field study in a university cafeteria found as small reductions in distance as 25 cm to result in 9–13% greater consumption of salads [20], the overall impact of placement strategies appears dependent on the magnitude of manipulation [14]. With minor manipulations that cause trivial changes in convenience and accessibility, effects on food choices may remain negligible [13]. For example, placement on the top versus bottom shelf of an 89 cm high display at the checkout counter had no effect on snack sales in a hospital canteen [28]. On the contrary, the selection of targeted foods increased significantly in a military dining facility along with changes to cafeteria layout that brought healthy options on more prominent and convenient places [25].

### 4.3. Perceived Influences on Food Choices

Our results regarding perceived influences on food choices demonstrate the multitude of factors individuals consider when choosing foods. Sensory appeal and healthiness seem to drive people’s food choices across cultures and populations [27,67,78,80]. The importance of familiarity, in turn, was likely pronounced because the cafeteria was a habitual food choice context for the majority of our participants, and because we grouped factors that reflected habitual choices to the familiarity domain.

Regarding behaviour change interventions, habitual environments have advantages and disadvantages. While consistent contexts and recurring behaviours provide fruitful elements for forming new habits, they can also strengthen already established habits and make them more resistant to change regardless of motivation and intentions [81,82]. Similar to many choice-architectural interventions, habits work through automatic, often unconscious and uncontrollable cognitive processes that mediate the effects of contextual cues on behaviour [10]. The shared working mechanism has raised a question about the capability of choice-architectural interventions to override habitual food choices [78]. Emerging evidence suggests that habits may indeed create barriers to the effectiveness of choice-architectural interventions [79]. This issue might concern particularly cognitively oriented interventions such as nutrition labels, as they require visual attention and aim to influence what people know [9]. Consumers have reported greater interest in nutrition labels and greater likelihood of using the labels when they buy products for the first time and when their need for nutrition information is higher [22,30]. On the contrary, the effects of interventions that influence behaviour more directly, even without noticing, might be more immune to established routines. Such behaviourally oriented interventions include, for example, default options and alterations to portion or tableware size [9].

Closely related to habits, many of our participants expressed detailed preferences or principles that guided their food choices. According to a recent review, people with strong preferences may be least susceptible to the effects of choice-architectural interventions [31]. Similarly, priming literature suggests that the effects of health-related primes on healthy choices could be dependent on the liking of targeted foods [15]. Supporting these claims, a field study found the use of nutrition labels more likely among individuals who are open to change and less bound to familiar meal choices [26]. These findings suggest that efforts to enhance the nutrition of individuals with strong preferences should employ strategies that target their preferred foods, for example, with gradual improvements to nutrient composition.

While evidence suggests that health primes and nutrition labels work for people with healthy preferences and intentions to eat healthy food [15,17,26,29,42], our results indicate that people may ignore such health-related cues despite health motivations. Potential explanations are many. First, people may consider foods served in workplace cafeterias generally healthy. This conception might reduce the need to seek for additional nutritional information [30]. Compared to meals in fast-food and full-service restaurants, meals in workplace cafeterias have proven to contain less energy [83]. Relatedly, eating in workplace cafeterias has been associated with healthier dietary habits [84,85,86]. Second, for many participants of our study, health-related motives did not focus on individual foods but rather targeted meal composition or remained less-specified higher-level goals. In addition, for several of our participants, healthiness appeared a relatively less important factor compared to sensory appeal. Prior research suggests that compared to specific health goals such as an attempt to reduce salt intake, general health goals may be too vague to trigger healthier food choices, particularly when challenged with competing motives such as taste [72,77] or hedonism [32]. A third remark relates to compensation. Our interviews indicated that participants could compensate the selection of less healthy food items by including or omitting other meal components or by regulating portion sizes. Such compensatory behaviours illustrate how the making of healthy choices can take various forms.

### 4.4. Strengths and Limitations

The strengths of this study included the real-world setting that guaranteed high external validity of study outcomes, the mixed-methods approach that drew a rich, multidimensional view of the studied phenomenon, and the study design that involved a control condition. Moreover, we demonstrated that eye tracking is a feasible data collection method in a natural cafeteria setting, and developed a method for analysing eye-tracking data collected in this context. This method enabled us to verify that the intervention was prominent enough to catch the eye, and allowed a systematic and rigorous tracking of intervention effects on visual attention and food choices. Eye-tracking outcomes were complemented by food-consumption data that provided objective evidence on the volume of foods consumed at the cafeteria level. Interviews, in turn, increased our understanding of the study population and supported the interpretation of eye-tracking and consumption results. The adopted mixed-methods design serves as an example of ways to combine objective, technology-driven data with self-reports to obtain more accurate, reliable, and meaningful outcomes than would be possible with any of the methods alone [87]. Additionally, the design answers a call for studies that examine the effects of nutrition labels on visual attention and food choices in real-world settings, considering person- and context-related factors [42].

The main limitation of this study is its small size. The small sample might lack statistical power to demonstrate significant effects even if they existed, particularly if true effect sizes are small and inter-individual dispersions large. For a larger sample, we would have needed a cafeteria with a larger customer base or several smaller cafeterias, longer data-collection period, and/or multiple eye-tracking glasses. Our study cafeteria served approximately 150 customers per day, but a substantial proportion of the clientele were regular visitors that ate in the cafeteria several times per week; thus, limiting the number of individuals that were eligible to participate during each study condition. During both control and intervention, recruiting new participants proved increasingly challenging towards the end of the week because customers keen to participate had already taken part, and customers unwilling to participate remained uninterested. In addition, most customers visited the cafeteria during a one-hour window from 11 a.m. to 12 p.m. With one pair of eye-tracking glasses, we could have only one participant at a time and were hence unable to make use of the peak hours. Including several cafeterias or extending the data collection period were not feasible, however, due to labour-intensive data-collection and -analysis methods. Resource issues are characteristic to mixed-methods studies that produce large volumes of data [48] and to studies that collect technology-driven data that need manual data-handling processes [46,87].

Another limitation of this study is its short duration, which may have influenced our findings because repeated exposures to nutrition labels are expected to enhance their noticing, understanding, and impact [70]. However, the label we used was familiar and understood, and the eye-tracking data demonstrated that the labels were seen. We thus doubt that a longer intervention would have substantially changed the results.

The study population in this study represented predominantly working population who valued food healthiness. Considering the location of the cafeteria in an office building, we assume that the majority of participants were office workers, who additionally may have represented a relatively highly educated and healthy-eating share of the workforce. In Finland, workers with higher education more commonly use workplace cafeterias compared to workers with lower education [4], and the use of workplace cafeterias is associated with healthier dietary habits [84,85,86]. The study cafeteria, in turn, likely had an offering with a relatively high nutritional quality—compared to full-service and fast-food restaurants at least [83]. Our results may not generalise to other occupational groups with different food choice motives or to other types of restaurants with diverse food offering.

When interpreting the outcomes of this study, a few methodological matters warrant consideration. Regarding eye tracking, we encourage keeping in mind that eye trackers are not mind-reading machines but produce approximate estimates of visual attention and cognitive processing. In a real-world setting, factors that can influence visual attention are myriad. For example, a queue at the serving line may have forced participants to kill time by viewing available foods, even without any intention to choose them. On the other hand, due to the reflexive tendency of eyes to follow sensed motion, participants’ gazes may have been drawn to foods that other customers were portioning, regardless of participants’ interest in these foods. Another remark relates to the accuracy of the eye-tracking measurement. In mobile eye tracking, the distance between participants and gazed objects varies and often differs from the distance used in calibration. This may compromise calibration accuracy and reduce the reliability of results [33]. A further consideration pertains to the proneness of manual data handling to researcher-originated errors [46]. In the present study, this issue concerns all collected data. Despite repeated and careful checks at all phases, the risk of random errors is evident due to the substantial manual work that our data collection, management, and analysis required. This uncertainty, however, concerns control and intervention data equally.

### 4.5. Recommendations

While nutrition labels typically receive support from the public [11,22,27,30,37,71] and in principle allow consumers to make informed healthy choices, we should not expect them to automatically trigger healthier eating. The labels and the nutritional criteria they build upon might be greater incentives for food manufacturers to improve the nutritional quality of food products [12,17], particularly when label use is mandatory. Similarly, labelling schemes could serve as standards for public food procurement. To increase healthier food choices, we recommend combining measures that ease the recognition and enhance the visibility of recommended options with measures that are less reliant on the provision of information and reflective cognitive processing, since such measures tend to yield greater effects [9]. Acknowledging the decisive influence that sensory appeal and habits have on food choices, efforts appear advisable that improve the nutritional profile of foods consumers prefer and that increase the attractiveness of foods with high nutritional quality. Regarding placement interventions, we encourage measures that substantially reduce the physical effort that healthy choices require. Additionally, in line with prior literature [79,88], we recommend future research to design interventions in collaboration with cafeteria staff, management, and clientele. Such approach facilitates the identification of factors that drive target groups’ food choices and the development of feasible interventions that tap into these factors. For multi-dimensional, more complete and meaningful effects evaluations of choice-architectural interventions, we recommend mixed-methods designs that combine objective and subjective measurements. Interventions that work through eyesight could benefit from eye-tracking measurements because they enable detecting the capabilities of interventions to capture visual attention, and allow monitoring food choices more accurately and reliably than self-reports or cashier data do. Future studies could follow the procedure developed in the current study to confirm our findings in different types of restaurants with diverse populations. To ensure larger study samples, researchers should strive for recruiting restaurants with large customer bases.

## 5. Conclusions

This study employed a mixed-methods approach and evaluated the effects of a real-world choice-architecture intervention that promoted nutritionally beneficial foods in a workplace cafeteria with priming posters, point-of-choice nutrition labels, and enhanced product placement. Additionally, the study developed a method for analysing eye-tracking data collected in a natural choice setting. The intervention proved capable of capturing visual attention to the posters and labels, yet ineffective in increasing healthier food choices or consumption among working-age consumers who prioritised sensory appeal and had established food-choice routines. While it is important to provide people with nutrition information in a quick-to-read and easy-to-grasp form, researchers, policy-makers, and practitioners should acknowledge the limited impact such information has on people’s food choices. To boost the effectiveness of health messages and visibility enhancements, we recommend complementing interventions with components that (1) address the determinants of target populations’ food choices, (2) enhance the sensory attractiveness of nutritionally favourable options, and (3) improve the nutritional quality of popular foods.

## Figures and Tables

**Figure 1 nutrients-14-03731-f001:**
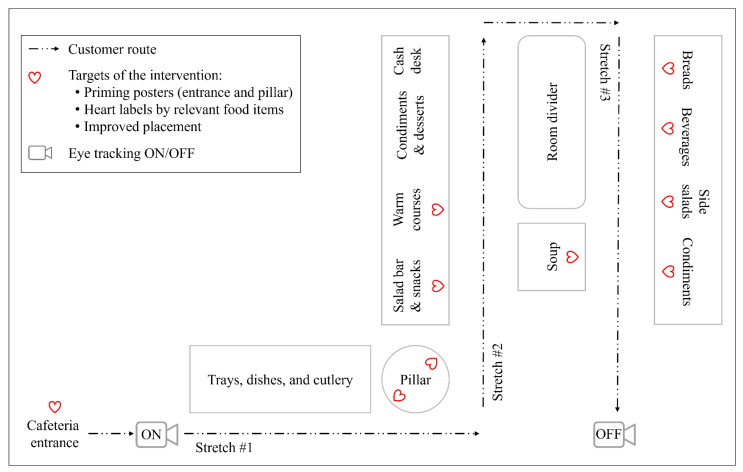
A schematic floor plan of the cafeteria serving line.

**Figure 2 nutrients-14-03731-f002:**
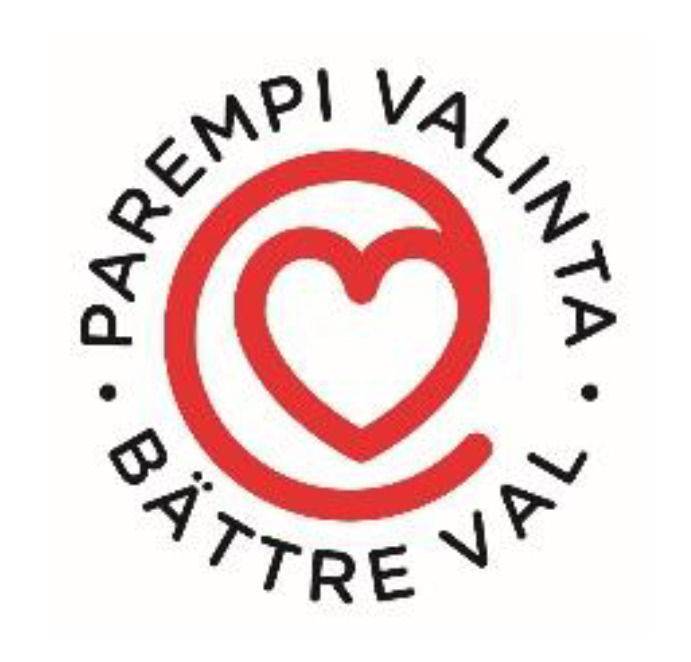
The Heart symbol with the text “better choice” in Finnish and Swedish. Image reproduced with the permission of the Finnish Heart Association.

**Figure 3 nutrients-14-03731-f003:**
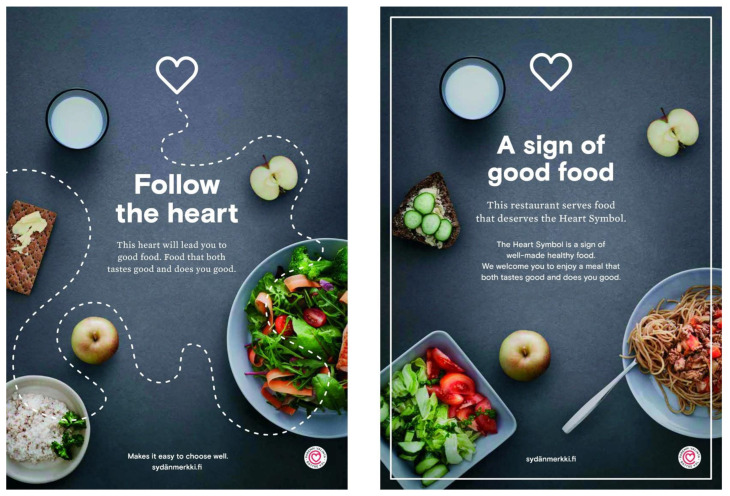
Posters that primed customers to notice and choose Heart-labelled foods. Original posters were in Finnish. Images reproduced with the permission of the Finnish Heart Association.

**Figure 4 nutrients-14-03731-f004:**
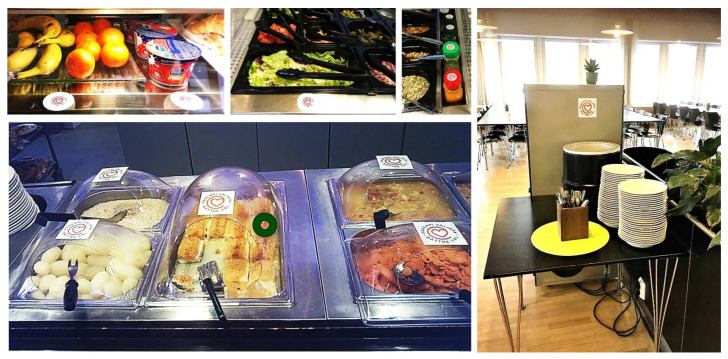
Examples of labelled heart-food items during the intervention. The dark green circle on the bottom left image indicates the point of a participant’s fixation. Images reproduced with the permission of the study cafeteria.

**Figure 5 nutrients-14-03731-f005:**
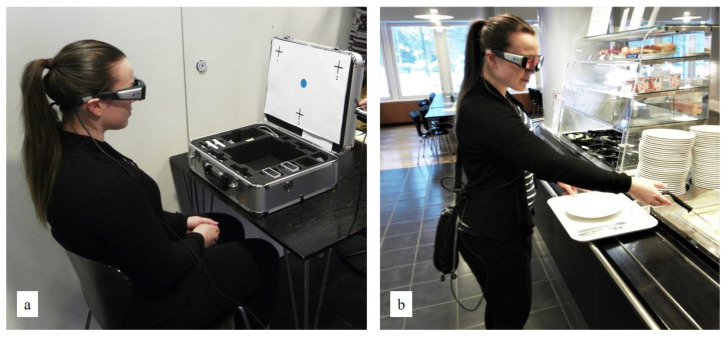
(**a**) The calibration of eye-tracking glasses; (**b**) Test subject wearing eye tracking glasses. Images reproduced with the permission of the study cafeteria and the test subject.

**Figure 6 nutrients-14-03731-f006:**
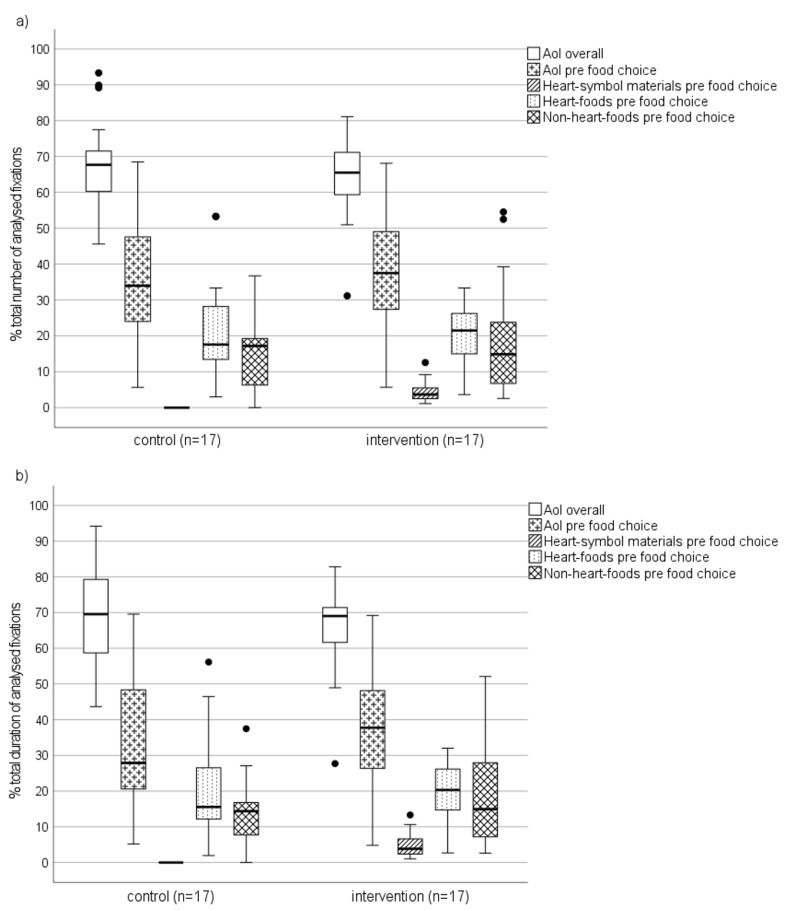
The distribution of (**a**) total number and (**b**) total duration of fixations on areas of interest (AoI, i.e., Heart-symbol materials, heart-foods, and/or non-heart-foods) as the percentages of total fixations accumulated at the analysed section of serving line stretch #2. Boxes extend from first to third quartile, horizontal lines across the boxes represent medians, whisker endpoints indicate minimum and maximum values, and dots represent outliers.

**Figure 7 nutrients-14-03731-f007:**
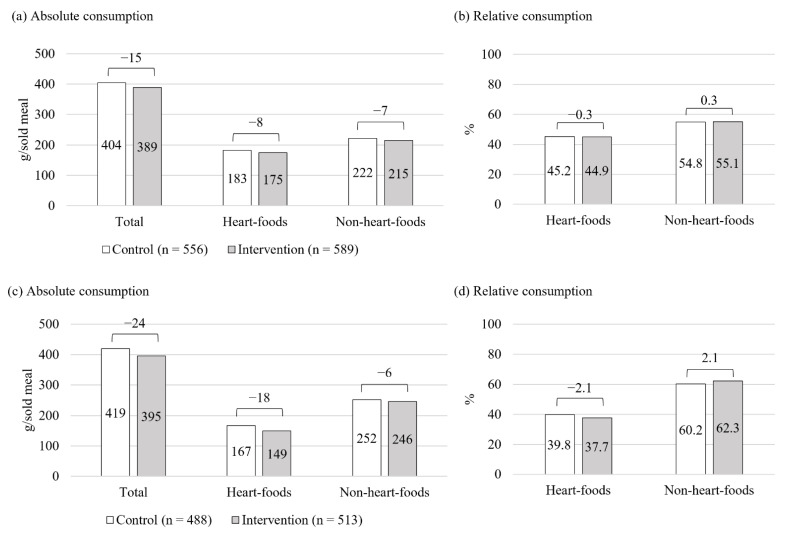
The cafeteria-level consumption of foods available on serving line stretch #2. (**a**,**b**) Data covers salad bar items, warm courses, and condiments; (**c**,**d**) Data covers warm courses and condiments. In graphs (**b**,**d**), numbers above the square brackets denote differences in percentage points.

**Table 1 nutrients-14-03731-t001:** The number of observations included in each analysis conducted in the study.

Analysis	Data	*n* Control + Intervention
Fixations on Heart-symbol materials and foods	Eye tracking	17 + 17 ^1^
Food choices	Eye tracking	22 + 19 ^2^
Perceived influences on food choices	Interview	22 + 19 ^2^
Self-reported observations and understanding of the Heart symbol	Interview	22 + 19 ^2^
Cafeteria-level food consumption	Food consumption	556 + 589 meals sold

^1^ Excludes participants with unsuccessful eye tracking or zero fixations on defined areas of interest before food choices (control: *n* = 5, intervention: *n* = 2; see Section 2.6.1) and includes four individuals who participated in eye tracking and interviews in both study conditions. ^2^ Includes seven individuals who participated in eye tracking and interviews in both study conditions.

**Table 2 nutrients-14-03731-t002:** Choice-architectural elements in place (x) during control and intervention. Heart-food refers to food items that met the product category-specific nutrition criteria of the Heart symbol.

Element	Description	Control	Intervention
Standard nutrition information	Heart-food main course options indicated with tiny black-and-white Heart symbols (font size ~8 pt.) next to allergen information on menu boards (size A4) at the cafeteria entrance and on the serving line.	x	x ^1^
	Pre-packaged heart-food items such as salad dressings featured small front-of-pack or back-of-pack Heart symbols. Seeing the symbols required lifting the products up from the serving line and reviewing product information.	x	x ^1^
Priming	Heart-foods promoted with posters (size A4–A3) at the cafeteria entrance and on two sides of a pillar at the end of serving line stretch #1. Each poster featured one of two slogans: “Follow the heart” or “A sign of good food”.		x
Prominent point-of-choice nutrition labels	Heart-foods and salad-bar notices (size A4) labelled with up to 10 × 10 cm Heart symbols on the serving line.		x
Placement	Heart-foods placed first in line and towards the front row, non-heart-foods last in line and towards the back row within product categories (i.e., snacks, salad components, salad dressings, warm courses, breads, and beverages). Heart-food snack options lifted at the eye level.		x

^1^ No changes were made to the information that was available already at baseline.

**Table 3 nutrients-14-03731-t003:** The total number and total duration of fixations on Heart-symbol materials, heart-foods, and non-heart-foods as the percentages of total fixations on areas of interest before food choices at serving line stretch #2. Control *n* = 17, intervention *n* = 17.

Fixation Target	Control			Intervention			Difference		
	Mean ^1^	SD	Range	Mean ^1^	SD	Range	Mean	95% CI	*p* ^2^
Heart-symbol materials									
% *n*	na	na	na	12.89	7.46	3.77–27.27	na	na	na
% duration	na	na	na	13.48	7.39	4.18–27.86	na	na	na
Heart-foods									
% *n*	60.79	23.14	11.76–100	57.81	20.02	16.67–85.00	2.98	−12.88, 18.85	0.702
% duration	60.68	23.53	9.52–100	57.47	20.00	21.74–87.75	3.21	−13.06, 19.49	0.688
Non-heart-foods									
% *n*	41.96	23.38	0.00–88.24	47.60	23.16	12.50–100	−5.63	−22.06, 10.79	0.486
% duration	42.31	23.92	0.00–90.48	48.27	23.47	10.67–100	−5.95	−22.82, 10.92	0.474

^1^ Percentages do not add up to 100% because the coding of fixations was not mutually exclusive. ^2^ Partially overlapping samples *t*-test “*T*_new1_” with equal variances assumed [53,55]. *p*-values < 0.05 are defined as statistically significant. All reported *p*-values are two-tailed. na = not applicable because targeted Heart-symbol materials were in place only during intervention.

**Table 4 nutrients-14-03731-t004:** The food items chosen at serving line stretch #2. All participants: control *n* = 22, intervention *n* = 19. Without salad-bar choosers: control *n* = 20, intervention *n* = 16.

Food Items Chosen	Control			Intervention			Difference
	Median	IQR	Range	Median	IQR	Range	*p* ^1^
All participants							
Total *n*	3	2	1–10	3	3	1–13	0.941
Heart-foods *n*	1	3	0–9	1	2	0–13	0.582
Heart-foods % total	33.3	78.8	0–100	66.7	75.0	0–100	0.261
Non-heart-foods *n*	1	1	0–4	1	2	0–3	0.163
Non-heart-foods % total	66.7	78.8	0–100	33.3	75.0	0–100	0.261
Without salad-bar choosers							
Total *n*	3	3	1–5	3	3	1–4	0.540
Heart-foods *n*	1	3	0–3	1	2	0–3	0.846
Heart-foods % total	33.3	72.9	0–100	50.0	68.8	0–100	0.366
Non-heart-foods *n*	1.5	1	0–4	1	1	0–3	0.314
Non-heart-foods % total	66.7	72.9	0–100	50.0	68.8	0–100	0.366

^1^ Partially overlapping samples “*T*_RNK1_” test for non-normally distributed variables with equal variances assumed [54]. *p*-values < 0.05 are defined as statistically significant. All reported *p*-values are two-tailed. IQR = interquartile range

**Table 5 nutrients-14-03731-t005:** The number (%) of individuals that mentioned each perceived influence on food choices. Control *n* = 22, intervention *n* = 19, total *n* = 34 ^1^.

Influence	Control	Intervention	Total
Sensory appeal	17 (77)	11 (58)	25 (74)
Healthiness	13 (59)	9 (47)	17 (50)
Familiarity	12 (55)	8 (42)	17 (50)
Particular foods	10 (45)	9 (47)	17 (50)
Variation	6 (27)	5 (26)	11 (32)
Weight control	6 (27)	3 (16)	9 (26)
Menu	8 (36)	1 (5)	9 (26)
Satiety	6 (27)	3 (16)	7 (21)
Mood	4 (18)	4 (21)	7 (21)
Special diet	2 (9)	4 (21)	6 (18)
Food quality	3 (14)	1 (5)	4 (12)
Convenience	2 (2)	2 (11)	4 (12)
Price	1 (5)	2 (11)	3 (9)
Season	2 (9)	1 (5)	3 (9)
Social influence	1 (5)	1 (5)	2 (6)
Natural content	1 (5)	1 (5)	2 (6)
Ethical concern	1 (5)	0 (0)	1 (3)

^1^ Individuals who participated both during control and during intervention and who mentioned the same influence on both times are counted in only once.

## Data Availability

The data presented in this study, including quantitative data and qualitative Finnish language data, will be made available on a reasonable request from the corresponding author.

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
