# Peer review of "Sensory Appeal and Routines Beat Health Messages and Visibility Enhancements: Mixed-Methods Analysis of a Choice-Architecture Intervention in a Workplace Cafeteria"

_nutrients, 2022, doi:10.3390/nu14183731_

Round 1
Reviewer 1 Report
The paper is describing a very innovative approach and study field – conducting and experimenting food choice in real life context. Despite the fact that I have a general major remark, I think authors have described the experimental set-up very well and knowing how tough it is, to conduct real-life experiments, scientific audience can very much learn and progress from this piece of work. Thanks for compiling this.
The main major remark/open question I have is:
Why did you conduct the study in this canteen? If I understood correctly, it is that this canteen was already certified to offer healthier foods, right? So, it not only had a certification + consultation scheme behind (which should be described, if this was the case), but also customers knew about the special quality.
I would very much assume that your effects are not strong, due to the fact, that customers already expected the ‘heart’-quality.
In this regard you should:
· Describe what the heart-food-cafeteria system is
· Describe what food they offer in regard to other canteens
· Describe (review) what food is usually consumed
· Describe WHO (which customers) frequently visit the canteen.
· Discuss the results in this regard and add this to the limitation
Further more, I would advise to argue/expect a discussion and outlook (recommendation for future studies)
· Why the sample size is so small
· How did you ensure that stretch 1 and 2 are okay, whereas 3 was not suitable? Please explain the quality criteria applied and your decision on using them.
Minors:
All through the document there are references (linked) missing. This should be taken care of.
Nevertheless, it is a sound and very sophisticated approach, you should definitely think about an upcoming study to conduct and take the learnings from this. And, having the above considered, this already unveils the huge amount of work and innovative approach you have designed.
Author Response
Point 1: The paper is describing a very innovative approach and study field – conducting and experimenting food choice in real life context. Despite the fact that I have a general major remark, I think authors have described the experimental set-up very well and knowing how tough it is, to conduct real-life experiments, scientific audience can very much learn and progress from this piece of work. Thanks for compiling this.
Response 1: We thank the reviewer for this positive feedback.
Point 2: Why did you conduct the study in this canteen? If I understood correctly, it is that this canteen was already certified to offer healthier foods, right? So, it not only had a certification + consultation scheme behind (which should be described, if this was the case), but also customers knew about the special quality.
I would very much assume that your effects are not strong, due to the fact, that customers already expected the ‘heart’-quality.
In this regard you should:
- Describe what the heart-food-cafeteria system is
- Describe what food they offer in regard to other canteens
- Describe (review) what food is usually consumed
- Describe WHO (which customers) frequently visit the canteen.
- Discuss the results in this regard and add this to the limitation
Response 2: These are very relevant questions indeed. In the following paragraphs, we try to clarify all the above-mentioned points and indicate corresponding sections in the manuscript.
Why did you conduct the study in this canteen? The main reasons for choosing the cafeteria were as follows:
- The cafeteria provided at least some foods that met the nutrition criteria of the Heart symbol besides foods that did not meet the criteria. The Heart symbol criteria build on national nutrition recommendations and acknowledge major public health nutrition challenges prevalent in Finland, including higher than recommended intake of salt and saturated fat, and lower than recommended intake of fibre. These nutritional challenges pertain to the working population as well.
- The cafeteria had standardised recipes with calculations of the nutritional content of all foods prepared in the kitchen (which not all workplace cafeterias have), and knowledge on which foods in their offering met and did not meet the Heart symbol criteria.
- The cafeteria had the right to label their criteria-fulfilling foods with the Heart symbol because they had applied and received this certificate from the Finnish Heart Association.
- The cafeteria made no efforts to inform customers about the heart symbol nor to indicate healthier options on the serving line.
These factors justified and ensured a feasible starting point for the implementation of our intervention that aimed to increase nutritionally high-quality food choices with choice-architectural strategies that facilitate the recognition and enhance the visibility of such choices. Intervention strategies suitable for the purpose included nutrition labels and related communication materials. The Heart symbol, in turn, was an appropriate labelling system because it has evidence-based, food category-specific nutrition criteria that provided a methodologically sound framework for the labelling intervention (https://www.sydanmerkki.fi/en/criteria/).
- We revised Section 2.2 L147─173 and section 2.4 L224-228 and L241-243 accordingly.
Did customers know about the special quality? Heart-symbol cafeterias are encouraged to inform their customers about the Heart symbol and to clearly indicate corresponding options on the serving line, but in our study cafeteria all this information was missing. At baseline, the cafeteria used no communication materials to inform about the Heart symbol and no point-of-choice labels that would distinguish healthier options from the less healthy options on the serving line. Hence, while the study cafeteria had the certificate and offered Heart-symbol options, the customers had no way of knowing this merely by observing the cafeteria environment. Thus, for the customers, the cafeteria appeared a typical Finnish workplace cafeteria. Moreover, during the preparation of the intervention, we discussed with the HR and the health and safety representatives of the municipal office that operated in the same building with the cafeteria, and learned that they were unaware of the fact that their workplace cafeteria was a Heart-symbol cafeteria. Therefore, considering the overall invisibility of the Heart symbol in the cafeteria environment and the lack of knowledge that our contacts in the office had regarding the Heart-symbol certificate, we assume that few customers were aware that they were eating in a Heart-symbol cafeteria. Moreover, even if some customers were aware of the certificate, nutritionally better options were nevertheless “invisible” because no efforts were made to facilitate their recognition or to enhance their visibility on the serving line. Hence, the study cafeteria provided an appropriate setting to study the effects of the used choice-architectural strategies while the food offering remained unchanged.
- We complemented Section 2.2 L165─172 accordingly.
Nevertheless, regardless of the customers’ awareness or unawareness of the Heart-symbol certificate, we agree with the reviewer that our findings may partly be explained by customers’ expectations regarding the quality of food served at workplace cafeterias. We have discussed this in section 4.3 as follows: “…people may consider foods served in workplace cafeterias generally healthy. This conception might reduce the need to seek for additional nutritional information [30]. Compared to meals in fast-food and full-service restaurants, meals in workplace cafeterias have proven to contain less energy [83]. Relatedly, eating in workplace cafeterias has been associated with healthier dietary habits [84–86]“. (L780─784)
- What the heart-food-cafeteria system is? Cafeterias that apply for the Heart symbol certificate from the Finnish Heart Association go through an application process that involves screening of the nutritional quality of the cafeteria’s food offering and identification of foods in the offering that meet relevant Heart-symbol criteria. When joining the system, cafeterias pledge to offer at least one meal option—including the main course, side dishes, bread, fat spread, milk/sour milk, fresh and cooked vegetables, and salad dressing—that fulfils the category-specific Heart-symbol criteria (https://www.sydanmerkki.fi/en/criteria-for-healthy-lunch/). Ideally, such a meal option would be available daily. Yet, cafeterias are still allowed to also offer options that do not meet the criteria, and this was the case in our study cafeteria. The cafeteria provided daily both options that fulfilled the criteria and options that did not.
- We complemented section 2.2 L147─165 accordingly.
- What food they offer in regard to other canteens? As portrayed in point (a), the study cafeteria provided daily at least one option in all meal components that met the category-specific nutrition criteria of the Heart symbol. The Supplementary table 1 that is enclosed with the manuscript details the food offering in the cafeteria during the study periods and the categorisation of available foods into heart- and non-heart-items. The offering comprised food items that workplace cafeterias in Finland commonly provide at lunch.
- We complemented section 2.2 L138 and L143─144 accordingly.
More generally, a recent multi-country study [83] that we refer to in the discussion (L782─783] found that in Finland, frequently purchased meals in workplace cafeterias contained less energy than frequently purchased meals in full-service and fast-food restaurants. Thus, the offering in our study cafeteria likely had a higher nutritional quality compared to the offering in nearby full-service or fast-food restaurants. Unfortunately, though, we are unaware of any more specific research that would have compared Heart-symbol cafeterias with other cafeterias in Finland regarding their offering and its nutritional quality. This, however, would be an interesting topic for future research.
[83] Roberts, S.B.; Das, S.K.; Suen, V.M.M.; Pihlajamäki, J.; Kuriyan, R.; Steiner-Asiedu, M.; Taetzsch, A.; Anderson, A.K.; Silver, R.E.; Barger, K.; et al. Measured Energy Content of Frequently Purchased Restaurant Meals: Multi-Country Cross Sectional Study. BMJ 2018, 363, k4864, doi:10.1136/bmj.k4864.
- What food is usually consumed? Cross-sectional population surveys that we refer to in section 4.3 (L783─784) have reported that within the working population of Finland, eating in workplace cafeterias is associated with healthier dietary habits, for example, in terms of vegetable and fish consumption [84–86]. Hence, we can assume that the customers of our study cafeteria represented a share of the working population that follows a relatively healthy diet. Thus, our findings may not generalise to workers who do not eat in workplace cafeterias.
[84] Raulio, S.; Roos, E.; Ovaskainen, M.-L.; Prättälä, R. Food Use and Nutrient Intake at Worksite Canteen or in Packed Lunches at Work among Finnish Employees. Journal of Foodservice 2009, 20, 330–341, doi:10.1111/j.1748-0159.2009.00157.x.
[85] Raulio, S.; Roos, E.; Prättälä, R. School and Workplace Meals Promote Healthy Food Habits. Public Health Nutrition 2010, 13, 987–992, doi:10.1017/S1368980010001199.
[86] Roos, E.; Sarlio-Lähteenkorva, S.; Lallukka, T. Having Lunch at a Staff Canteen Is Associated with Recommended Food Habits. Public Health Nutrition 2004, 7, 53–61, doi:10.1079/PHN2003511.
Regarding cafeteria-level food consumption, our findings indicated that heart-foods covered approximately 45 % of the overall consumption during both study conditions (L552─554). Comparing this finding to other studies at workplace cafeterias is problematic, however, because the methodology, setting, food offering, and categorisation of foods into healthy and less healthy are not comparable to other studies.
- WHO (which customers) frequently visit the canteen? Unfortunately, we do not have information on the customers who visited the cafeteria frequently because we did not measure the frequency at which the participants visited the cafeteria. This was out of the scope of the current study. Overall, we have limited information on the characteristics of the clientele because we did not collect identifiable data on the study participants. What we now is that in general, the customers represented predominantly working population, a substantial proportion of whom worked in the municipal office that operated in the same building with the cafeteria (L130─132). Based on this knowledge, we can assume that the customers were predominantly office workers. They may additionally have represented a relatively highly educated share of the workforce, because in Finland, workers with higher education more commonly use workplace cafeterias compared to workers with lower education [4] (L837─841). Additionally, we know on a general level that a substantial proportion of the clientele were regular customers (L216─222). Yet, we have no means of identifying these customers accurately.
[4] Valsta, L.; Kaartinen, N.; Tapanainen, H.; Männistö, S.; Sääksjärvi, K. Ravitsemus Suomessa - FinRavinto 2017 -Tutkimus. Nutrition in Finland - The National FinDiet 2017 Survey; Report 12/2018; National Institute for Health and Welfare: Helsinki, Finland; ISBN 978-952-343-238-3.
- Discuss the results in this regard and add this to the limitation. In the above responses, we have indicated the lines of the manuscript that discuss each point. Additionally, in accordance with the responses, we complemented the limitations section 4.4. as follows: “The study population in this study represented predominantly working population who valued food healthiness. Considering the location of the cafeteria in an office building, we assume that the majority of participants were office workers, who additionally may have represented a relatively highly educated and healthy-eating share of the workforce. In Finland, workers with higher education more commonly use workplace cafeterias compared to workers with lower education [4], and the use of workplace cafeterias is associated with healthier dietary habits [84–86]. The study cafeteria, in turn, likely had an offering with a relatively high nutritional quality—compared to full-service and fast-food restaurants at least [83]. Our results may not generalise to other occupational groups with different food choice motives or to other types of restaurants with diverse food offering.” (L836─845).
Point 3: Furthermore, I would advise to argue/expect a discussion and outlook (recommendation for future studies)
Response 3: We complemented the recommendations section 4.5 as follows: “Future studies could follow the procedure developed in the current study to confirm our findings in different types of restaurants with diverse populations. To ensure larger study samples, researchers should strive for recruiting restaurants with large customer bases.” (L886─889)
Point 4: Why the sample size is so small
Response: Factors that limited the sample size included the size of the cafeteria’s customer base, the duration of the daily lunch service, and the availability of merely one pair of eye-tracking glasses. The cafeteria served approximately 150 customers per day (L130), but a substantial proportion of the clientele were regular visitors that ate in the cafeteria several times per week. Thus, the total number of individuals who visited the cafeteria was far less than 5x150 per workweek; limiting the number of individuals that were eligible to participate during each study condition. The cafeteria served lunch daily from 10.30 a.m. to 1 p.m. (L132─133), but during this 2.5-hour time window, the customer flow was uneven, as most customers came between 11 a.m. and 12 p.m. With one pair of eye-tracking glasses, we could have only one participant at a time and were hence unable to make use of the peak hours.
- We complemented the limitations section 4.4 (L815─828) accordingly.
Point 5: How did you ensure that stretch 1 and 2 are okay, whereas 3 was not suitable? Please explain the quality criteria applied and your decision on using them.
Response 5: The first author was responsible for the implementation, made needed adjustments daily before the beginning of the lunch service, and monitored the quality of implementation throughout the intervention. On serving line stretches #1─2, the implementation followed plans as defined in Table 2 (L241─243) except for minor weaknesses in the placement of a couple of warm courses during two days of the intervention. According to a literature review on the implementation of prevention and health promotion interventions, expecting perfect or near-perfect implementation is unrealistic and unnecessary, because few interventions have reached implementation levels closer than 80 % of optimal and because studies have yielded positive results with levels around 60 % [56]. We hence considered the implementation on stretches #1─2 overall satisfactory. On serving line stretch #3, however, the implementation faced major challenges throughout the intervention week because most food items kept travelling away from their assigned places and corresponding Heart labels as customers handled them. Such implementation quality was unacceptable, as the findings would not have reflected the intended intervention. Hence, we chose to exclude the stretch #3 from the analyses.
[56] Durlak, J.A.; DuPre, E.P. Implementation Matters: A Review of Research on the Influence of Implementation on Program Outcomes and the Factors Affecting Implementation. Am J Community Psychol 2008, 41, 327–350, doi:10.1007/s10464-008-9165-0.
- We complemented Section 2.4 (L230─233 and L252─269) accordingly.
Point 6: All through the document there are references (linked) missing. This should be taken care of.
Response 6: Thank you for pointing out these technical flaws. The cross-links have been corrected.
Point 7: Nevertheless, it is a sound and very sophisticated approach, you should definitely think about an upcoming study to conduct and take the learnings from this. And, having the above considered, this already unveils the huge amount of work and innovative approach you have designed.
Response 7: We thank the reviewer for this encouraging evaluation.
Reviewer 2 Report
Review comments for Sensory appeal and routines beat health messages and visibility enhancements: mixed-methods analysis of a choice-architecture intervention in workplace cafeteria
The paper describes using a unique approach of priming posters, point-of-choice nutrition labels, and product placement, attempting to influence consumer food choices in a workplace cafeteria. The mixed method, quantitative and qualitative, provides significant information on consumer behavior changes.
There are many missing references and broken words (e.g., line 256, the “w” and “ith”) throughout the paper. The cross-references of the tables or figures in the results section are missing. Please revise.
Specific comments:
· Lines 135-137, The cafeteria provided daily four warm main course options: two fish/meat courses, one vegetarian course, and one soup,…..
Suggest expanding the description of the daily menu with an example. Were both healthy and unhealthy food items offered daily for the main courses, breads, or salad dressings as a comparison of choices?
· Lines 173-174, As a result, the samples recruited during control and during intervention included partly same and partly different individuals.
How many participants were in both control and intervention? Suggest authors providing a diagram of study design with the number of participants in the control and intervention of the study.
· Line 254, .. that take 30 samples per second….
Do authors mean that the eye-tracking glasses take 30 photos per second?
· Line 269, …(section 0).
According to Figure 1 on page 21, where is “section 0”?
· Lines 324-325, …(control: n=3581, intervention: n=3680) from 37 participants with unbroken eye-tracking recordings (control: n=19, intervention: n=18).
Compared to lines 178-179, authors described that there were 41 customers (control: n=22, intervention: n=19) participated in eye-tracking and interview. The numbers of participants are not clear. Suggest authors providing a diagram of study design with the number of participants in the control and intervention phases of the study.
· Lines 340, 341, …(Supplementary table 1)….. (Supplementary table 2).
Please provide the Supplementary tables 1 and 2 for the paper.
· Line 426, … , excluding snacks and desserts due to incomplete data.
Can authors expand the condition of “incomplete data”? Was it due to missing in the data collection or coding error?
· Line 429, … or if this was not possible, removed the food item from the data.
Can authors expand the condition of “not possible”? what type of food item were removed from the data? Are they the heart-foods or non-heart foods?
· Cafeteria-level food consumption section 2.6.3
Although many studies consider total volume of food sold as the proxy measurement for the food consumption. Can authors provide some insights on the observation of differences of food consumption for heart-foods and non-heart foods?
· Line 555, Individuals that participated both during control and during intervention are counted in only once.
Do authors mean that when individuals provided the same influence factor in both control and intervention phases of the study, the same mentioned factor was only counted in once? Please clarify.
· Lines 814-815, For example, a queue at the serving line may have forced participants to kill time by viewing available foods, even without any intention to choose them.
Can authors provide additional information regarding the serving line condition? Was it rushing the people through the serving line due to the crowded condition during mealtime?